# Using Recurrent Procedures in Adaptive Control System for Identify the Model Parameters of the Moving Vessel on the Cross Slipway

**Hanna Rudakova, Oksana Polyvoda and Anton Omelchuk *** 

Engineering Cybernetics Department, Kherson National Technical University, Kherson 73008, Ukraine; RudakovaAnna25@gmail.com (H.R.); pov81@ukr.net (O.P.)

* Correspondence: tareon@ukr.net; Tel.: +380-500317468

**Abstract:** The article analyses the problems connected with ensuring the coordinated operation of slipway drives that arise during the launch of a ship. The dynamic model of load of the electric drive of the ship's cart is obtained taking into account the peculiarities of the construction of the ship-lifting complex, which allows us to analyse the influence of external factors and random influences during the entire process of launching the ship. A linearized mathematical model of the dynamics of a complex vessel movement in the process of descent in the space of states is developed, which allows us to identify the mode of operation of the multi-drive system, taking into account its structure. The analysis of application efficiency of recurrent methods for identification (stochastic approximation and least squares) of the linearized model parameters in the space of states is carried out. A decision support system has been developed in the automated system of operational control by the module for estimating the situation and the control synthesis to ensure a coherent motion of a complex ship-carts object in a two-phase environment.

**Keywords:** the slipway; the adaptive control system; the least squares method; the periodic procedure; the parameters estimation; a large-sized object; a recurrent algorithm; the identification procedure

## 1. Introduction

Shipbuilding and ship repairing are the most important sectors of the economy in countries with access to the sea or which have large river systems. Depending on the technology of the construction of vessels, as well as their size and purpose, different facilities for moving ships to and from the water are used [1–3].

The slipways make up about 30% of all existing boat launch systems and apply to 60% of shipbuilding facilities. Most slipways, for example in the CIS countries, were built in the 60s and 80s of the twentieth century. The main problems of these facilities are the use of outdated control systems on the one hand and their considerable deterioration on the other, these facts a number of difficulties during the operation of the launching facilities. The ship-lifting facilities are complex electromechanical systems. For the operational control it is advisable to use the means of automated systems. The type of slipway complex is presented in Figure 1.

For descent or ascent using a slipway, the vessel is mounted on special carts, which moves along rail tracks is located on an inclined plane at an angle α of 6–10°. Each cart is driven by a single electric drive using a steel cable. Carts have the ability to independently move down along the rails down (up), while the main task, during the descent (lifting) of the vessel, is the coordinated movement of carts, in which the vessel should move uniformly at a given speed without distortions.

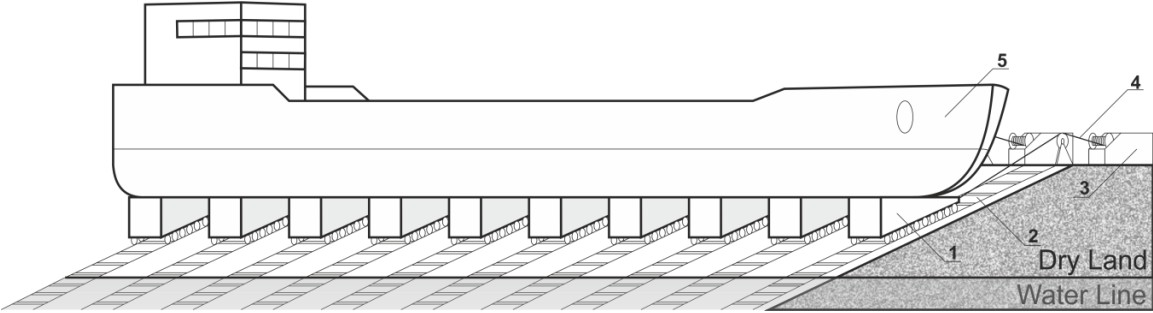

**Figure 1.** Cross slipway: 1—carts; 2—rail tracks; 3—electric drives; 4—cables; 5—vessel.

The process of launching the ship can be divided into several stages (Figure 2):

1. The movement of the cart with acceleration to the necessary constant mode speed of descent, $l \in [0, l_1]$. $l_0$—the distance from the starting point of the cart to the point of the cable stop.

2. The movement of the cart at a constant speed before entering the water, $l \in [l_1, l_2]$. $l_3 - l_2 = l_T$—appropriate cart width.

3. Transfer of the cart from the slipway surface to the underwater part, $l \in [l_2, l_3]$. $l_w$—distance to water line.

4. Full submersion carts under water, $l \in [l_3, l_4]$.

5. The movement of the cart at a constant speed in the water before the vessel ascends, $l \in [l_4, l_5]$.

6. The ascent of the ship and the braking of the cart to a full stop, $l \in [l_5, l_6]$.

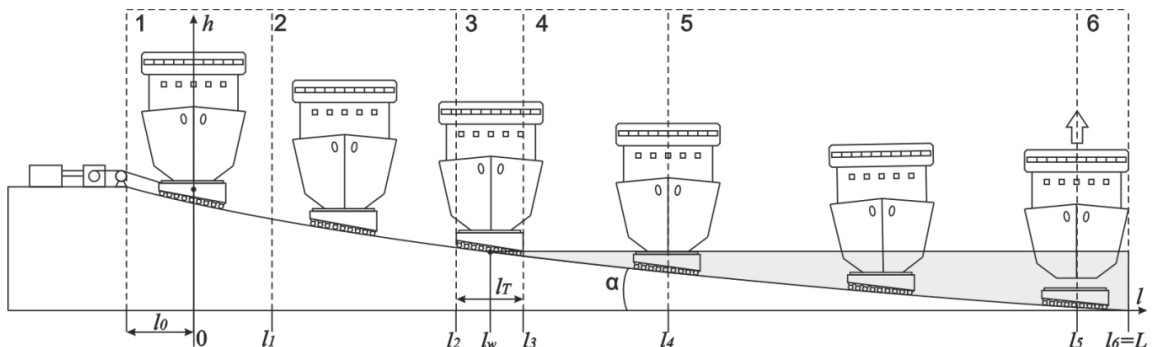

**Figure 2.** Stages of launching the vessel using a slipway.

Control of the motion parameters of the trigger car must be carried out throughout the journey.

Despite the existence of the regulatory rules for the technical operation of ship-lifting facilities, it is necessary describe the step by step the sequence of actions during the preparation and vessel launching, the emergency situations occurred periodically. The main reason for this is the uneven load on the electric drives, and, accordingly, the overload of electric motors and cables.

The overhaul of the ship-building facilities implies substantial financial costs, while the cost of modernizing the control system is several times lower.

The bulky objects motion is performed with help of the cooperative work of electric motors and mechanisms complex. Movement processes happen in non-stationary conditions when exposed to external effects that vary extensively in all points of movement, which frequently could create the nonstandard situations. The reliable operation of mentioned complexes could be done only with application of adaptive control systems taking in account the stochastically changing outer and inner factors. In the synthesis of rational management of the work of the components of electromechanical complexes, there periodically take place problems of identifying the parameters of the model of the motion process.

Lately, the artificial intelligence methods often used to execute the process identification, such as fuzzy logic, neural networks and so forth.

But to make use the neural network technology possible, it is required to employ a large amount learning set for a teaching of the artificial intelligent system, which is not always possible, due to the lack of a data set for training or limited training time [4,5].

While fuzzy systems application the design problem of base rules set and membership functions could occur connected with the heavy conditions caused by the rapid changing factors which have impact on the process.

## 2. Model Description

For a uniform movement of the launching carts, as well as in the areas of its acceleration and deceleration, it is necessary to constantly provide the appropriate cable tension. The main forces acting to the cart during the descent are shown in Figure 3.

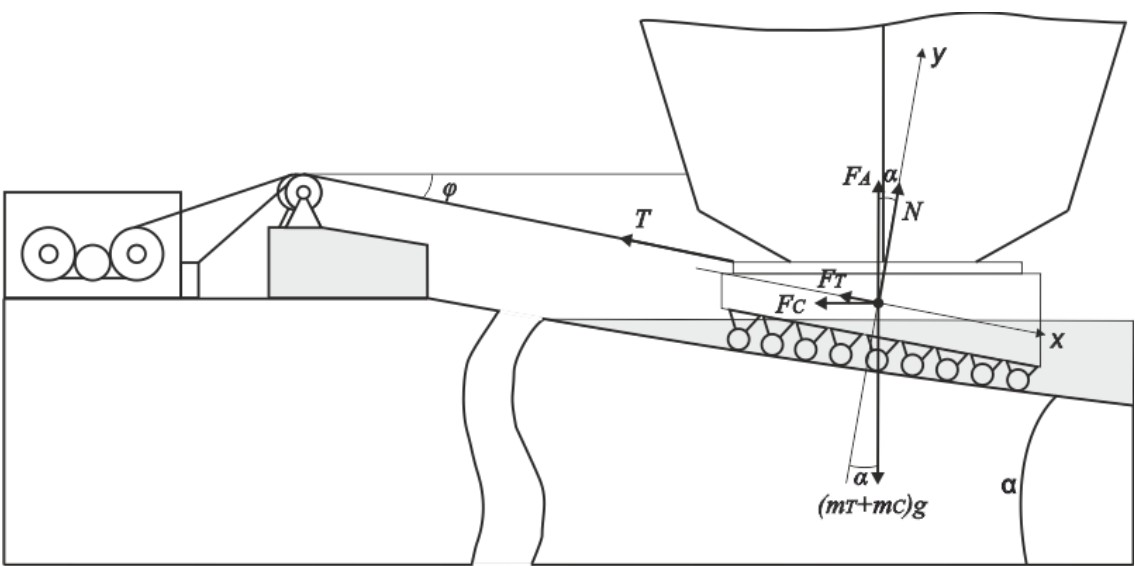

**Figure 3.** Forces that acting to the cart during the launch of the vessel.

Using the second law of Newton, we will create a system of equations for a cart with a ship mounted on it, taking into account all the forces acting on the cart along the entire path (both by land and in water):

$$\sum F_x = (m_T + m_C)g \cdot \sin(\alpha) - T \cdot \cos(\phi - \alpha) - F_T - F_C \cdot \cos(\alpha) - F_A \cdot \sin(\alpha) = (m_T + m_C)a,$$
$$\sum F_y = N + T \cdot \sin(\phi - \alpha) + F_A \cdot \cos(\alpha) - mg \cdot \cos(\alpha) - F_C \cdot \sin(\alpha) = 0, \tag{1}$$
$$F_T = \mu \cdot N,$$

where $m_T$ is the mass of the cart, $m_C$ is the mass of the ship section, $\alpha$ is the slope angle of the slip;

$T$—cable tension, $\phi$—the deflection of the cable;

$F_T$—friction force, $F_C$—water resistance force, $F_A$—Archimedes force;

$a$—cart acceleration, $\mu$ is the coefficient of rolling friction between the wheels of the cart and the rails, $N$ is the reaction force of the support.

Thus, from system (1) it is possible to express the tension of the cable $T$ in the following form

$$T = \frac{((m_T + m_C)g - F_A) \cdot (\sin(\alpha) - \mu \cdot \cos(\alpha)) - F_C(\mu \cdot \sin(\alpha) + \cos(\alpha))}{\mu \cdot \sin(\phi - \alpha) - \cos(\phi - \alpha)}. \tag{2}$$

The forces acting on the cart when moving in water (Archimedes force $F_A$ and the resistance force from the water side $F_C$) have the form defined by the following relations:

$$F_C = k_c \, S \frac{\rho v^2}{2}, \tag{3}$$

$$F_A = \rho g V_A, \tag{4}$$

where $k_c$ is the coefficient of resistance of the vessel and the cart (for the central sections of the vessel, can be taken $\approx 1$);

$S = S_C + S_T$ is the characteristic surface area of the vessel and cart, $v$—cart speed, $\rho$ is the density of water;

$V_A = V_C + V_T$ is the volume of the vessel and the cart submerged in water.

Consider the dependence of all changing factors on the distance travelled $l$. In Figure 4 shows the geometry of the object (slip), the ratio of its height and length.

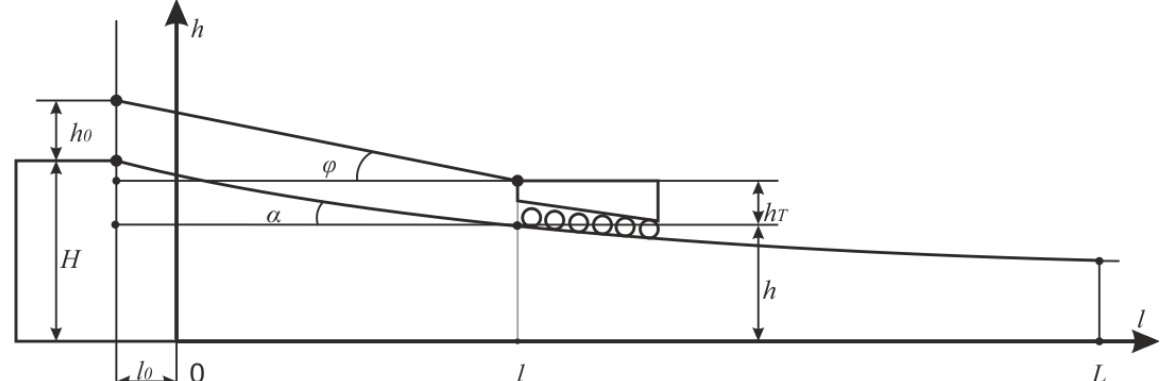

**Figure 4.** Graph of the angle of inclination of the cable $\phi$.

The change in the height of the slip is relative to its length and can be described by the exponential dependence $h(l) = H \cdot e^{-\lambda l}$ or, taking into account a small change in the angle of inclination—$\alpha \approx 4°$ from the slip geometry can be found as

$$h(l) = H - l \cdot sin(\alpha), \tag{5}$$

where $H$ is the height of the slipway.

Figure 4 shows the angle of inclination of the cable $\phi$ during the descent process without taking into account the extension of the cable.

From geometric relationships, we can get the following relation

$$\phi = arctg\left(\frac{(H - h) + (h_0 - h_T)}{l + l_0}\right), \tag{6}$$

where $h_0$ is the height of the block; $h_T$ is the height of the cart; $h$—the current height of the cart.

To take into account the impact on the object of Archimedes force, it is necessary to consider the change in the vessel and the vessel submerged in water $V_A$, since, according to (4), $F_A = f(V_\Pi)$. Due to the design features of the slipway, first, the cart will be submerged in water, which has a very small volume compared to the ship hull, then the lower, narrower part of the vessel enters the water; after that the volume $V_A$ will increase nonlinearly. Figure 5 shows the cart and section of the vessel when submerged under water and the dependence of the total submerged volume $V_A$ on $l$, where 1 is the surface part of the vessel and zones 3 and 2 are the submerged parts of the cart and vessel, respectively.

Figures 6–8 shows the different diagrams of submerged volume of the cart $V_T$ and the vessel section $V_C$, in dependencies from the traveled path $l$.

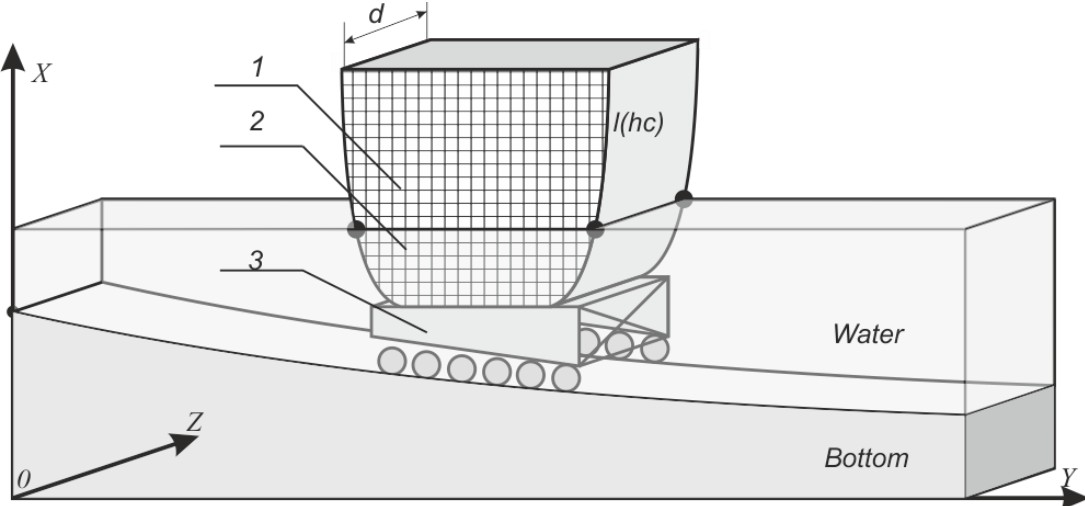

**Figure 5.** Visual cart and vessel section submerged volume change.

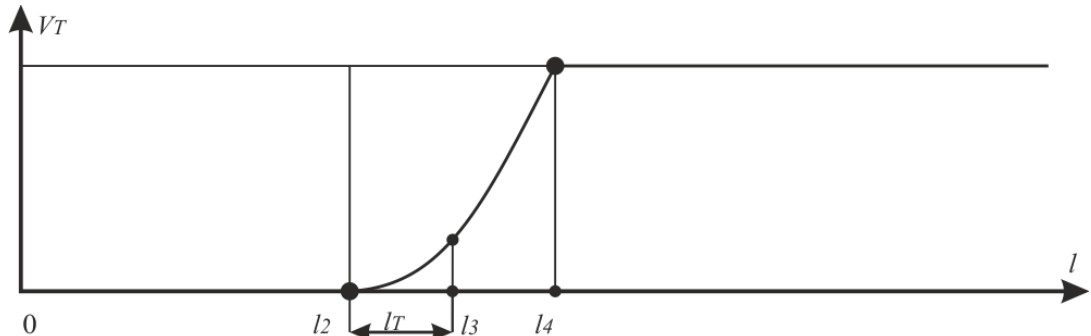

**Figure 6.** Diagram of cart submerged volume $V_T$ change.

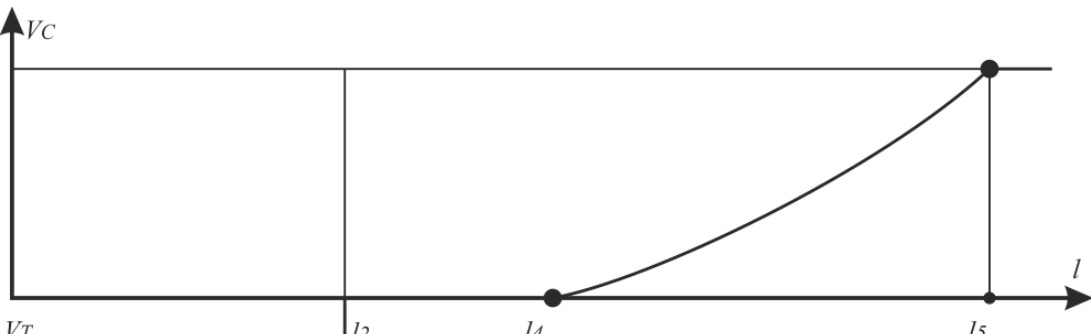

**Figure 7.** Diagram of typical vessel section submerged volume $V_C$ change.

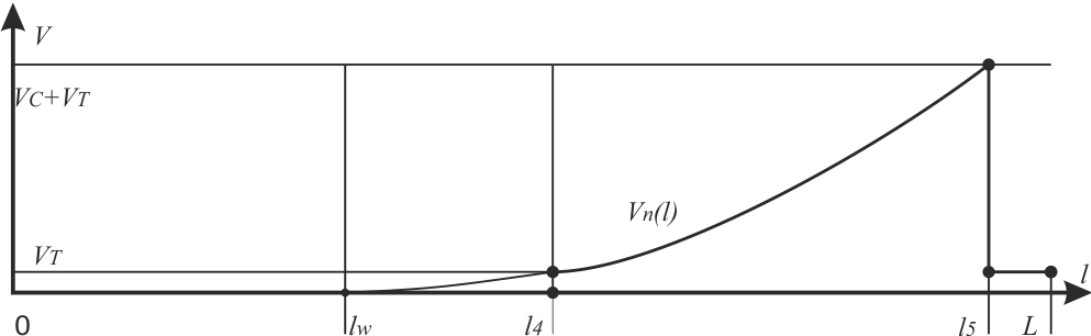

**Figure 8.** Diagram of cart and vessel section submerged volume change.

Thus, the impact of the buoyancy force can be predicted with the known law of volume change. Due to the fact that the volume of the launch of the submerged carriage is extremely small and the same for all carts, it can be neglected. The volume of the vessel section in Cartesian coordinates system can be calculated using the triple integral [6].

The characteristic area of the vessel is the area which is perpendicular to the flow of water. In this case, for the central part of the vessel, the characteristic area of the vessel is equal to the area of the rectangle formed by the length of section $d$ and the height of the vessel $h_v$. If to neglect the characteristic area of the cart $S_T$

$$S = h_v \cdot d. \tag{7}$$

The characteristic area $S$ has an effect on the force of water resistance and it becomes greater as the ship sinks deeper into the water, therefore, it depends on $l$. Figure 9 shows the dependence of $S$ on the slip length $l$.

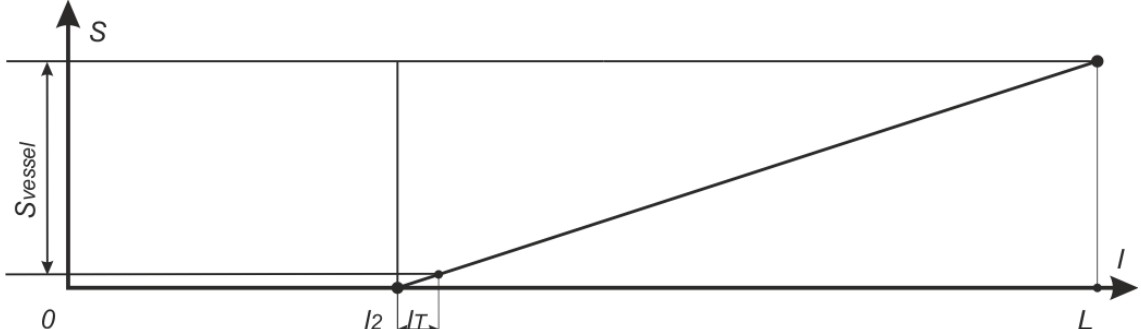

**Figure 9.** Changing the characteristic area of the vessel and carts while diving.

The coefficient of friction $\mu$ when driving over land or water has constant values. However, when a cart enters the water (zone 3), it smoothly changes (Figure 10).

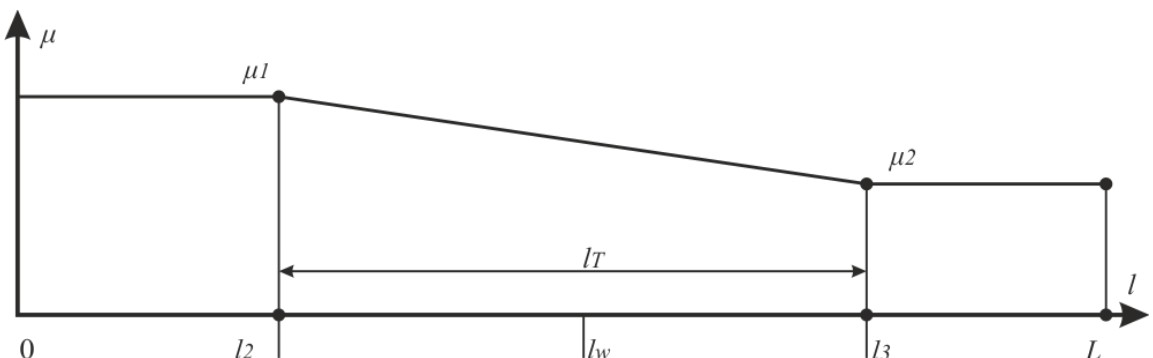

**Figure 10.** The change of the coefficient of rolling friction during immersion.

To describe the dependence of $\mu$ on the path traveled, you can use z-shaped membership functions [7], which leads to the expression

$$\mu(l) = \mu_2 + \frac{\mu_1 - \mu_2}{1 + e^{\delta \cdot (l - l_B)}}, \tag{8}$$

where $\mu_1$, $\mu_2$—coefficients of friction between the wheels of the cart and the rails on land and in water, respectively; $\delta$ is the coefficient of steepness.

The coefficient of slope $\delta$ is as follows

$$\delta = \frac{4}{l_T} \cdot ln\left(\frac{1}{\varepsilon} - 1\right), \tag{9}$$

where $\varepsilon$ is the admissible error.

Since the motion parameters in different zones are distinguished by the presence and magnitude of some forces and factors, for modelling it is necessary to set them accordingly, as shown above. Possible values of the parameters considered by zones are shown in Table 1.

**Table 1.** Movement parameters at various stages of descent.

| Parameters | Zones | | | | | |
|---|---|---|---|---|---|---|
| | 1 | 2 | 3 | 4 | 5 | 6 |
| Cart weight, $m_T$ | $m_1$ | $m_1$ | $m_1$ | $m_1$ | $m_1$ | $m_1$ |
| Ship section weight, $m_C$ | $m_2$ | $m_2$ | $m_2$ | $m_2$ | $m_2$ | 0 |
| Acceleration, $a$ | $a_1 = const,$ $a_1 > 0$ | 0 | 0 | 0 | 0 | $a_2 = const,$ $a_2 < 0$ |
| Slipway slope angle, $\alpha$ | | | | $\alpha(l)$ | | |
| Cable deflection angle, $\phi$ | | | | $\phi(l)$ | | |
| Coefficient of friction, $\mu$ | $\mu_1$ | $\mu_1$ | $\mu_1 \div \mu_2$ | $\mu_2$ | $\mu_2$ | $\mu_2$ |
| Cart area, $S_T$ | 0 | 0 | $0 \div S_T$ | $S_T$ | $S_T$ | $S_T$ |
| Ship section area, $S_C$ | 0 | 0 | 0 | 0 | $0 \div S_C$ | 0 |
| Cart volume, $V_T$ | 0 | 0 | $0 \div V_T$ | $V_T$ | $V_T$ | $V_T$ |
| Ship section volume, $V_C$ | 0 | 0 | 0 | 0 | $0 \div V_C$ | 0 |
| Speed, $v$ | $0 \div v_d$ | $v_d$ | $v_d$ | $v_d$ | $v_d$ | $v_d \div 0$ |

Given specific values of the parameters, it is possible to obtain for them analytical dependencies based on relations (2)–(9) and to analyse the changes in the loads (forces) and parameters of interest when the trigger car moves along the entire length of the slipway.

When analysing the movement of the trigger carriage, it is necessary to take into account the change of parameters that affect the process of descent at all stages—from the beginning of movement to the end.

The dependences and relations obtained allow us to determine the cable tension $T$ depending on the given mode of motion (speed $v_d$, acceleration $a_1$ and deceleration $a_2$) throughout the entire route. This makes it possible to estimate the load on the electric drive during the descent process.

To optimize (match) the drive control of a complex electromechanical system for launching a ship into the "slipway" type, it is advisable to develop a dynamic model of the engine of the launching carriage in the state space based on Equation (2).

The bulky object motion on a plane can be described in form of sums of translational and rotational motions, which are presented by expressions relatively to the centre of mass [7]:

- for translational motion

$$m\frac{dv}{dt} = \sum F, \frac{dl}{dt} = v, \tag{10}$$

- for rotational motion

$$J\frac{d\omega}{dt} = \sum M, \frac{d\phi}{dt} = \omega, \tag{11}$$

where $m$ is the mass of the object, $v$ is the translational motion speed, $l$ is the motion length, $t$ is the time, $\sum F$ is the total value of all external forces applied to the object, $J$ is the object inertia moment relative to the rotation axis, $\omega$ is the object angular velocity, $\phi$ is the object rotation angle, $\sum M$ is the total rotational moment of all forces relatively to rotation axis of the mass centre of a bulky object. A diagram for determining the parameters of motion for two points of measurement is shown in Figure 11.

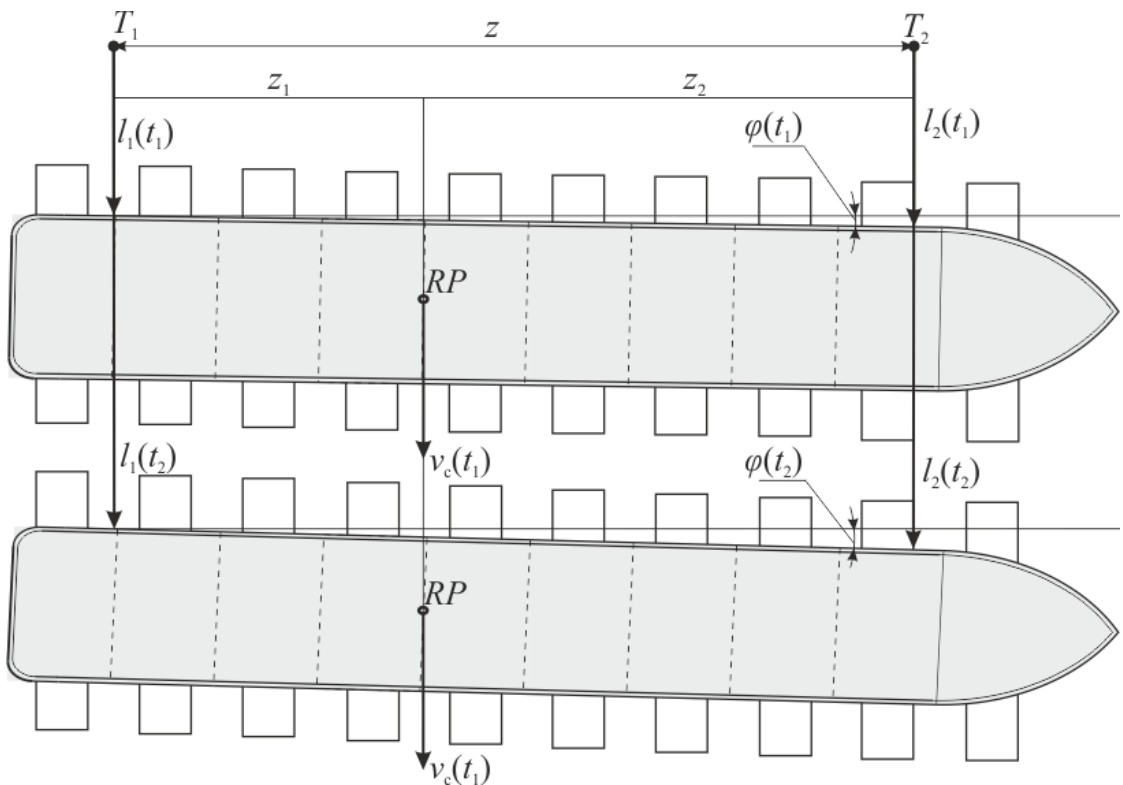

**Figure 11.** Scheme for the determining of the parameters of motion.

The system of relations (10), (11) can be written as expressions in the state space of the fourth order [8] with state vector of the object being moved $\mathbf{x} = \begin{pmatrix} x_1 & x_2 & x_3 & x_4 \end{pmatrix}^T$, where $x_1 = S$, $x_2 = v$, $v = dS/dt$ $x_3 = \phi$, $x_4 = \omega = d\phi/dt$; control actions vector $\mathbf{u}$ and output vector $\mathbf{y}$ describe the construction of the monitoring system. Usually, such systems model has non-linear relations.

While synthesis of optimal control actions, the methods of modern control theory the linearized model of system usually are used, presented in the state space [9].

To identify the parameters of the motion of a large object, which must be taken into account when controlling the mode of operation of the slipway and the synthesis of drive control, it is possible to measure the distance of some points of the object when moved from their original position.

One of the obvious solutions to the problem of ensuring a consistent movement of the launch trucks is to provide control over the position of the stern and bow of the vessel (Figure 12). Indeed, most of the problems arising from the launching of a ship on the track of the slip have a direct impact on the position of the ship. Due to the design of the slip, the misalignment of the movement of the carts (their position is not in an "even row") leads to a change in the position of the hull, its distortion and displacement.

Under slip operation conditions, the most acceptable technical characteristics are the use of optical location means, since it requires measurement in the open air in the range of 1–100 m ± 0.01. Given the shape of the hull of the vessel (Figure 13), it seems convenient to measure the distance to the hull using laser range finders.

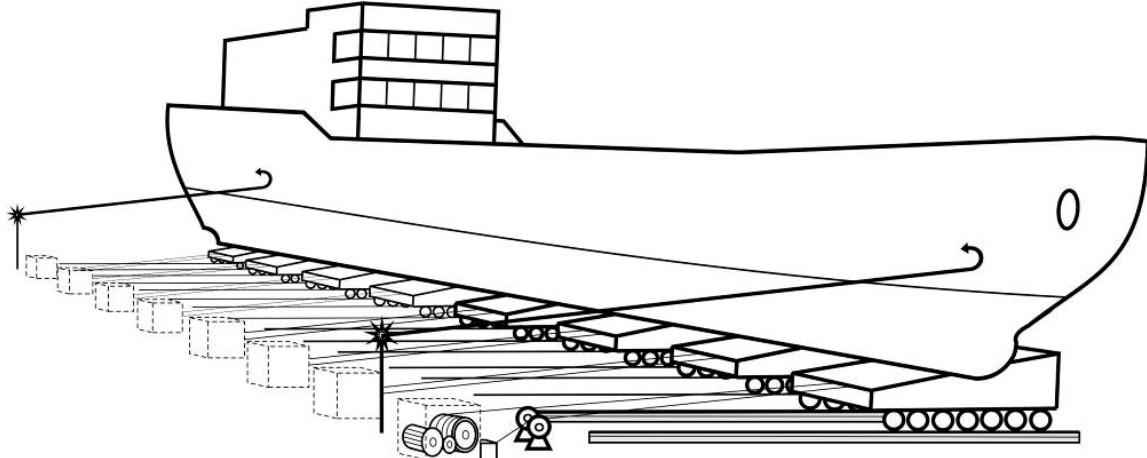

**Figure 12.** Control of the ship movement with a range finder—general view.

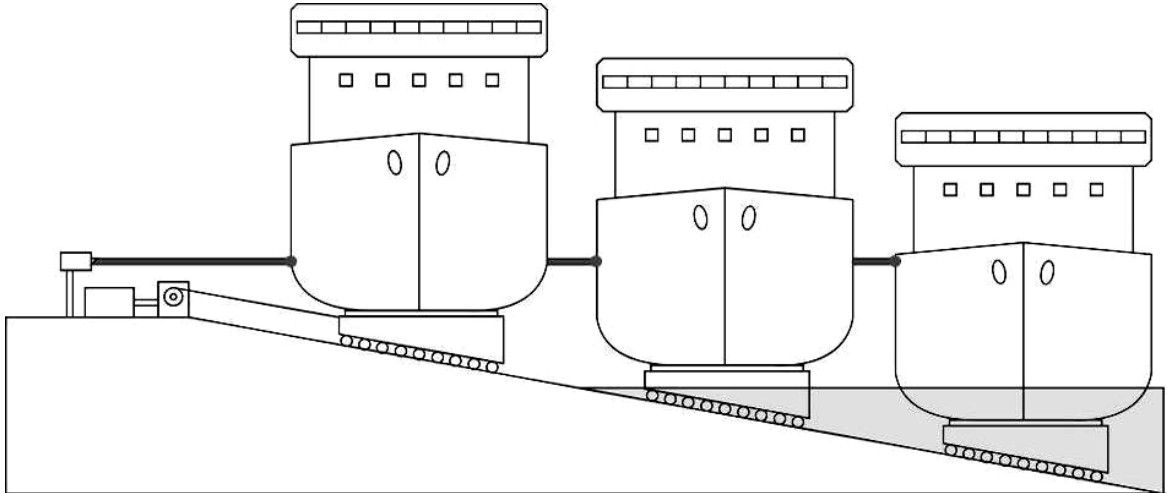

**Figure 13.** Control of ship movement with a range finder—side view.

Consider the process of moving a bulky object by example of a ship's descent on a slipway. The impacting forces on moving "ship-cart" distributed objects in time of a controlled downhill motion are presented in Figure 14.

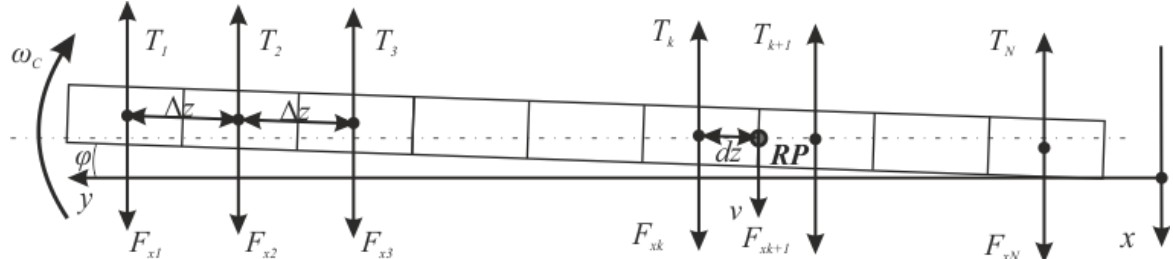

**Figure 14.** The structure of forces acting on the "ship-carts" object.

The ship motion model on a slipway, designed in the states space, which considers all the important outer influences, is nonlinear [10].

Based on the analysis of the equations of motion, the equations of state of the model of the ship-cart object are formulated in the state space in the form

$$
\begin{aligned}
\dot{x}_1 &= x_2, \\
\dot{x}_2 &= \frac{1}{m}\sum_{i=1}^{N} F_{xi}(x_1) - \frac{T_m}{m}\sum_{i=1}^{N} u_i, \\
\dot{x}_3 &= x_4, \\
\dot{x}_4 &= \frac{1}{J}\sum_{i=1}^{N}(T_m u_i - F_{xi}(x_1))\cdot[(k-i)\Delta z + dz]\cdot\cos x_3,
\end{aligned}
\tag{12}
$$

where $x_1 = l$ is the displacement of the center of mass of the vessel along the $x$ axis;

$x_2 = v$ is the rate of translational motion of the center of mass along the $x$ axis;

$x_3 = \phi$—the angle of rotation of the vessel;

$x_4 = \omega$—vessel rotation speed;

$\mathbf{u} = \begin{pmatrix} u_1 & u_2 & \dots & u_N \end{pmatrix}^T$ is the control vector, $u_i = T_i/T_m$, $i = \overline{1, N}$; $T_i$ is the tension force of the cable of the i-th cart;

$T_m$—maximum allowable cable tension; $F_{xi}(l_i)$—the load on the $i$-th drive, due to the forces affecting the cart at the point of the movement path $l_i$;

$m = m_c + N\cdot m_T$ is the mass of the vessel-carts object, $m_c$ is the mass of the vessel, $N$ is the number of carts, $m_T$ is the mass of the cart;

$J$ is the moment of inertia of the vessel;

$\Delta z$ is the distance between the centers of neighboring carts; $dz$ is the distance from the rotation point ($RP$) (centre of mass of the object) to the centre of the $k$-th cart.

The output equations of the model are

$$
y_j = x_1 - [(k-j)\Delta z + dz]\sin x_3, \ j = \overline{1, r},
\tag{13}
$$

where $r$ is the number of outputs needed to identify the motion parameters of a complex object.

The following mathematical model was gained in the states space in the vector-matrix form in the result of linearization

$$
\begin{aligned}
\dot{\hat{\mathbf{x}}} &= \mathbf{A}\hat{\mathbf{x}} + \mathbf{B}\hat{\mathbf{u}}, \\
\vec{\mathbf{y}} &= \mathbf{C}\cdot\vec{\mathbf{x}}
\end{aligned}
\tag{14}
$$

The matrices $\mathbf{A}$, $\mathbf{B}$ and $\mathbf{C}$ have a constant structure but a change depending on the operating conditions of the system and have the form

$$
\mathbf{A} = \begin{pmatrix} 0 & 1 & 0 & 0 \\ a_{21} & 0 & a_{23} & 0 \\ 0 & 0 & 0 & 1 \\ a_{41} & 0 & a_{43} & 0 \end{pmatrix}
$$

$$
\mathbf{B} = \begin{pmatrix} 0 & 0 & \cdots & 0 \\ K_1 & K_1 & \cdots & K_1 \\ 0 & 0 & \cdots & 0 \\ q_1 K_2 & q_2 K_2 & \cdots & q_N K_2 \end{pmatrix}
\tag{15}
$$

$$
\mathbf{C} = \begin{pmatrix} 1 & 0 & q_1 & 0 \\ 1 & 0 & q_N & 0 \end{pmatrix}
$$

The elements of the matrix $\mathbf{A}$ are defined as

$$
a_{21} = f_{21}(x_{1s}, x_{3s}), \ a_{23} = f_{23}(x_{1s}), \ a_{41} = f_{41}(x_{1s}, x_{3s}), \ a_{43} = f_{43}(x_{1s}),
\tag{16}
$$

where $x_{1s}, x_{3s}$ are the elements of the stable state vector of the object $\mathbf{x}_s = (x_{1s}, \dots, x_{4s})^T$ in a bounded neighborhood;

$K_1 = -T_m/m$, $K_2 = T_m/J$ are scale coefficients;
$q_i = (k - i)\Delta z + dz$, $i = \overline{1, N}$ is distance of the *i*-th force application point from the rotation point (the bulky object mass centre).

The elements of the matrix **C** correspond to the structure of the measurement system.

The block diagram of the adaptive control system of complex object with observer and regulator, is shown in Figure 15.

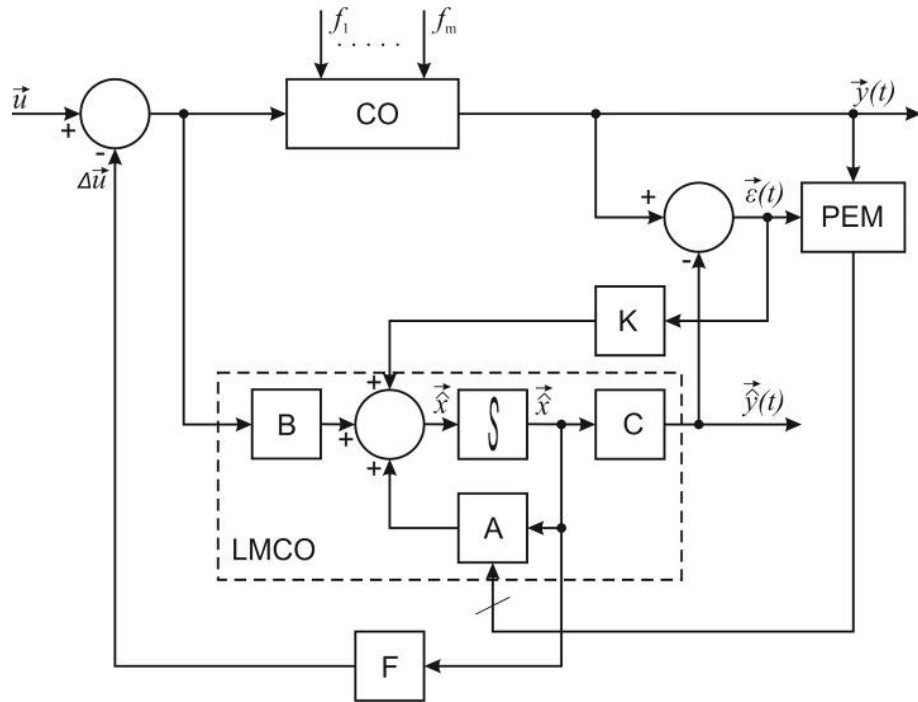

**Figure 15.** Block diagram of adaptive control system.

The Parameter Estimation Module is marked at Figure 15 as "PEM," which implements the recurrent algorithm for matrix identification **A** of the linearized model control object (LMCO). The values changes of the matrix **A** components lead to the need to search a new values of observer matrix K and regulator coefficients F and, accordingly, to solve of Riccati equation [11].

The elements of the matrices **A**, **B** and **C**, can change in time the object motion process relying on the conditions of the system operation, saving matrices structure.

The elements of the matrices, which depend on the outer and inner operating conditions of the system, may change, which causes the need of tuning the values of the model parameters to keep its adequacy. As a result, with the tuning of the motion trajectory parameters according to the outer conditions, it is need to start at intervals an operation for identifying model parameters.

To perform the motion process model parameters identification in real time, it is suitable to involve recurrent procedures which allow getting an estimate of the model parameters if new measurement results come up [11]. The next formula describes the recurrent evaluation procedures

$$\mathbf{P}[k+1] = \mathbf{P}[k] + \gamma[k+1] \cdot f(\mathbf{P}[k], \mathbf{y}[k+1], \mathbf{u}[k+1]) \tag{17}$$

where $\mathbf{P}[k]$ is the current evaluation of the parameter; $\gamma[k]$—the weight coefficient; $f$ is special function that relies on the current value $\mathbf{P}[k]$ and defines the size and trend of the following step; $\mathbf{y}[k+1]$ and $\mathbf{u}[k+1]$ are the signals (output and input) that go after the current value.

The stochastic approximation method and the least square method are the most known recurrent procedures [12]. Because the accuracy of the stochastic approximation method could only be gained

for $t \to \infty$, as described in Reference [11], so it cannot be implemented while performing model parameters identification for solving problems of controlling the bulky object motion in real time.

Let us consider identification procedures using the methods of recurrent stochastic approximation and least squares to determine the parameters of the model for the process of moving a ship on a slipway, represented in the form (17).

While getting new measuring data on the control object at discrete moments of time, it is convenient to implement models in form of the state space as following

$$\mathbf{x}_m[k] = \mathbf{A}[k-1] \cdot \mathbf{x}_m[k-1] + \mathbf{B} \cdot \mathbf{u}[k-1]. \tag{18}$$

The deviation values based on analysis of the state variables obtained from the model $\mathbf{x}_m[k]$ and as a result of measurements $\mathbf{x}_o[k]$ let evaluate the adequacy of the model to object, that is, by the value of the error

$$\mathbf{e}[k] = \mathbf{x}_o[k] - \mathbf{x}_m[k]. \tag{19}$$

The algorithm for setting parameters has the form

$$\mathbf{A}[k] = \mathbf{A}[k-1] + \mathbf{\Gamma}[k] \cdot \mathbf{e}[k] \cdot \mathbf{x}_m^T[k], \tag{20}$$

where $\mathbf{\Gamma}[k]$—is the matrix obtained on the basis of real values of the state vector measured over the entire observation interval $t \in \left[t_0, t_f\right]$, which can be determined recurrently by different ways.

In the case of the method of stochastic approximation as:

$$\mathbf{\Gamma}[k] = \mathbf{\Gamma}[k-1] - \mathbf{\Gamma}[k-1] \cdot \mathbf{x}_o[k] \cdot \gamma[k-1] \cdot \mathbf{x}_o^T[k] \cdot \mathbf{\Gamma}[k-1], \tag{21}$$

where

$$\gamma[k-1] = \left[1 + x_o^T[k] \cdot \mathbf{\Gamma}[k-1] \cdot \mathbf{x}_o[k]\right]^{-1} \tag{22}$$

In the case of the least squares method the matrix $\mathbf{\Gamma}[k]$ as:

$$\mathbf{\Gamma}[k] = \mathbf{I}_n \cdot \frac{\gamma}{k}, \gamma > 0, \tag{23}$$

To implement the both methods, it is necessary to specify the initial values of the state vector components of the model $\mathbf{x}_m[0] = \mathbf{x}_o[0]$ and also the matrix components initial values $\mathbf{A}[0]$, for the known dynamics of the control vectors u[k] and the state of the object $\mathbf{x}_o[k]$ for $k = 1, 2, 3, \ldots N$. The initial values of the matrix $\mathbf{\Gamma}[0]$ are chosen as $\mathbf{\Gamma}[0] = \mathbf{I} \cdot (1/\alpha)$, where $\alpha$ is a numerical coefficient whose value influence the convergence of the identification algorithm.

The identification phase ends when tiny deviations of the matrix **A** components values occur for all *i* and *j*, that is, the following conditions is satisfied:

$$\left|a_{ij}[k] - a_{ij}[k-1]\right| < \delta \tag{24}$$

Let us consider identification procedures using the recurrent methods of stochastic approximation and least squares to determine the parameters of the model of the process of moving a ship on a slip.

In the process of calculations using the recurrent method of stochastic approximation, the best results were obtained using the algorithm tuning coefficient $\gamma = 0.5$ and $\Delta t = 1$, for the initial values of the model state vector $\mathbf{x}_m[0] = \begin{bmatrix} 0 & 0.05 & 0 & 0 \end{bmatrix}^T$. The initial and final values of the components of the matrix **A** are as follows

$$\mathbf{A}[0] = \begin{bmatrix} 1 & 0 & 0 & 0 \\ 0 & 1 & 0 & 0 \\ 0 & 0 & 1 & 0 \\ 0 & 0 & 0 & 1 \end{bmatrix}, \mathbf{A}[300] = \begin{bmatrix} 1.061 & -0.651 & 0.064 & 2.016 \cdot 10^{-3} \\ 0.012 & 0.356 & -9.77 \cdot 10^{-4} & 4.78 \cdot 10^{-4} \\ 0.087 & -0.081 & 0.51 & 2.4 \cdot 10^{-4} \\ -2.03 \cdot 10^{-5} & 2.57 \cdot 10^{-4} & 1.7 \cdot 10^{-6} & 0.5 \end{bmatrix}. \tag{25}$$

The change in the values of the matrix **A** components in the calculation process is shown in Figure 16. The structure of the resulting matrices (25) is significantly different from the structure of the matrix, resulting from the linearization of the nonlinear motion model. The finite time of the identification phase using the recurrent stochastic approximation method was $t > 350$ s, which is unacceptably large when the vessel is moving, since the external conditions for the movement of a large object change with new objects and the definition of new matrix **A** values is required.

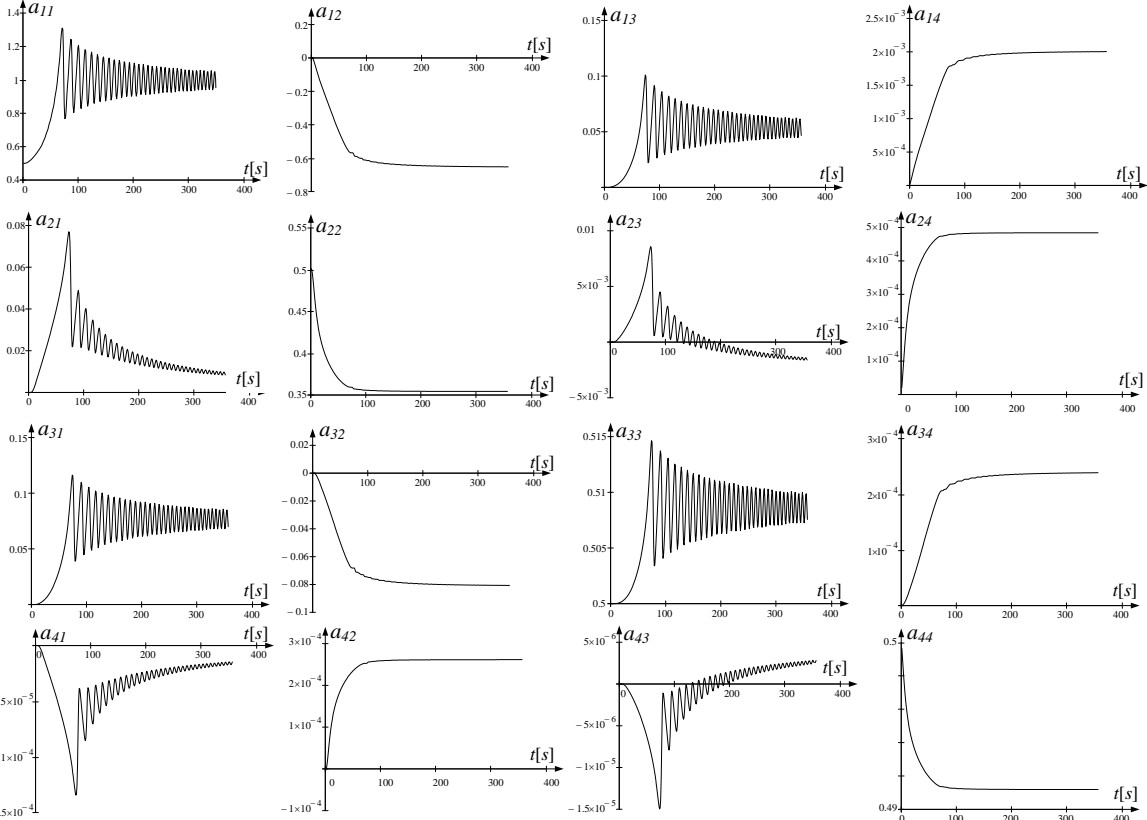

**Figure 16.** Dynamics of the adjustment of the values of the matrix **A** components obtained by using stochastic approximation method.

According to the theory [13,14], the accuracy of the stochastic approximation method can be considered only with $t \to \infty$ [9], therefore it is not advisable to use it when identifying model parameters to solve problems of controlling the movement of a large object in real time.

It is advisable to analyse the effectiveness of more accurate identification methods in the control system, which allow you to maintain the structure of the matrixes of the linearized model of the process of moving the vessel on the slipway, for example, the least squares method [15,16].

The identification process was modelled with application of the recurrent least squares method with the initial values of the object state vector and the initial values of the components of the matrix **A** of the form

$$\mathbf{x}_o[0] = \begin{bmatrix} 0 \\ 0.05 \\ 0 \\ 0 \end{bmatrix}, \quad \mathbf{A}[0] = \begin{bmatrix} 0 & 1 & 0 & 0 \\ 1 & 0 & 1 & 0 \\ 0 & 0 & 0 & 1 \\ 1 & 0 & 1 & 0 \end{bmatrix}. \tag{26}$$

The change in the values of the matrix A components during the calculations is shown in Figure 17.

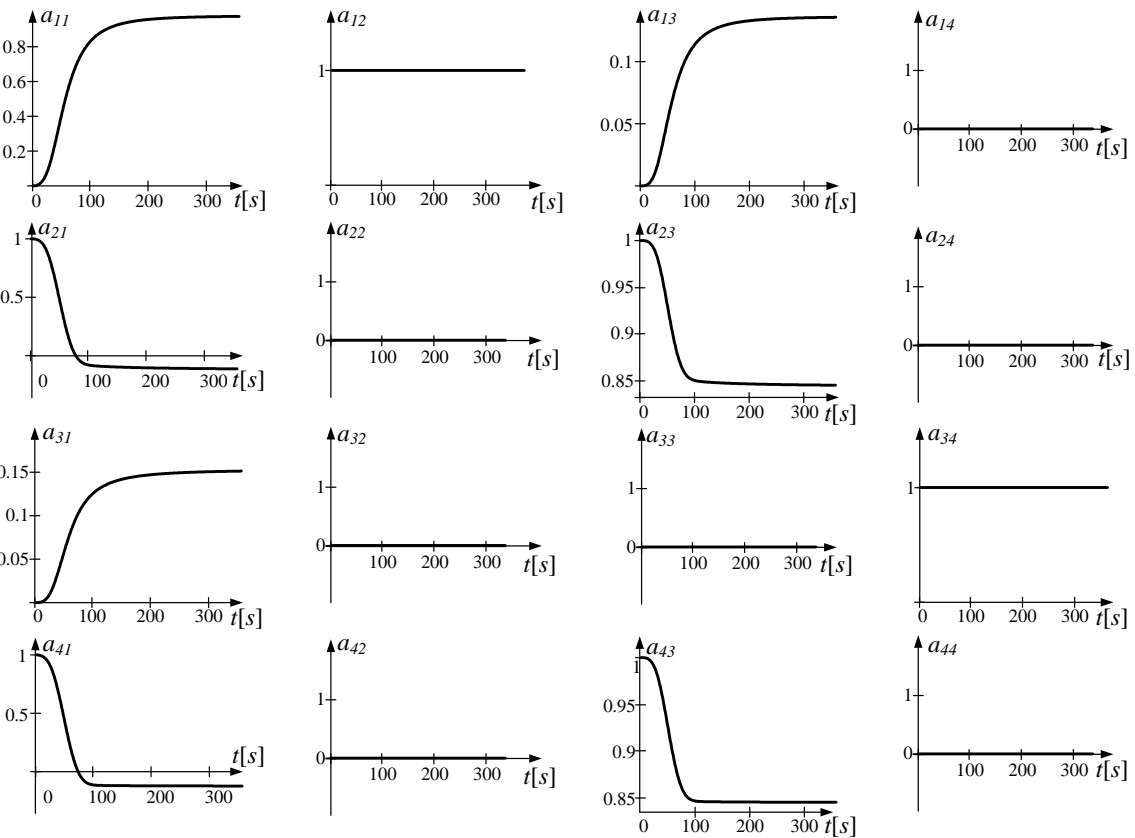

**Figure 17.** Dynamics of adjusting the values of matrix **A** components obtained by using least squares method.

As a result of the calculations, a matrix **A** of the following form obtained

$$\mathbf{A}[350] = \begin{bmatrix} 0.98 & 1.02 & 0.14 & 5.63 \cdot 10^{-5} \\ -0.12 & -0.03 & 0.85 & -6.75 \cdot 10^{-5} \\ 0.15 & 2.95 \cdot 10^{-3} & 0.02 & 1 \\ -0.13 & -0.03 & 0.85 & -7.22 \cdot 10^{-5} \end{bmatrix}. \tag{27}$$

Taking into account the permissible error of 5%, the resulting matrix **A** can be written as

$$\mathbf{A}^* = \begin{bmatrix} 0.98 & 1 & 0.14 & 0 \\ -0.12 & 0 & 0.85 & 0 \\ 0.15 & 0 & 0 & 1 \\ -0.13 & 0 & 0.85 & 0 \end{bmatrix}. \tag{28}$$

Setting values $\gamma = 150$ and $\Delta t = 1$ s gives the best results by using the algorithm. Using recurrent stochastic approximation method gives the time of identification phase $t = 100$ s, which is acceptable, because only with $t > 300$ s the outer conditions of the bulky object motion changes and the calculation of new matrix **A** values is need.

The graphs of the state vector components dynamics of the object (line 1) and the model (line 2) are shown in Figure 18.

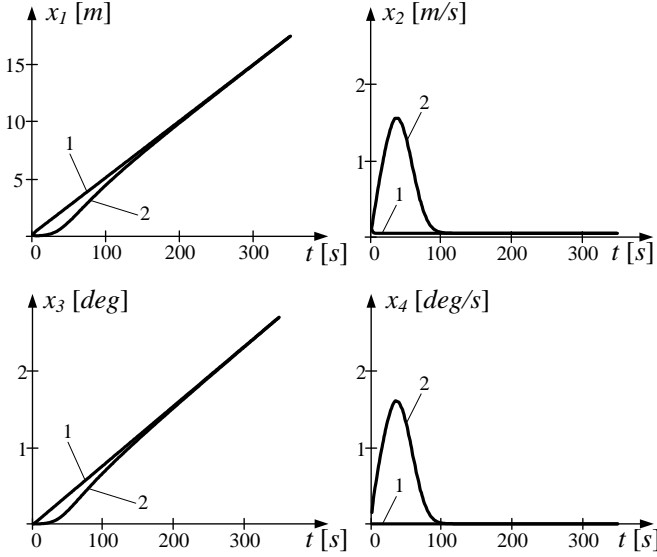

**Figure 18.** Graphs of dynamics of the states vector components.

Graphs of the error dependence $\Delta x_i(t) = x_o(t) - x_m(t)$ are shown in Figure 19.

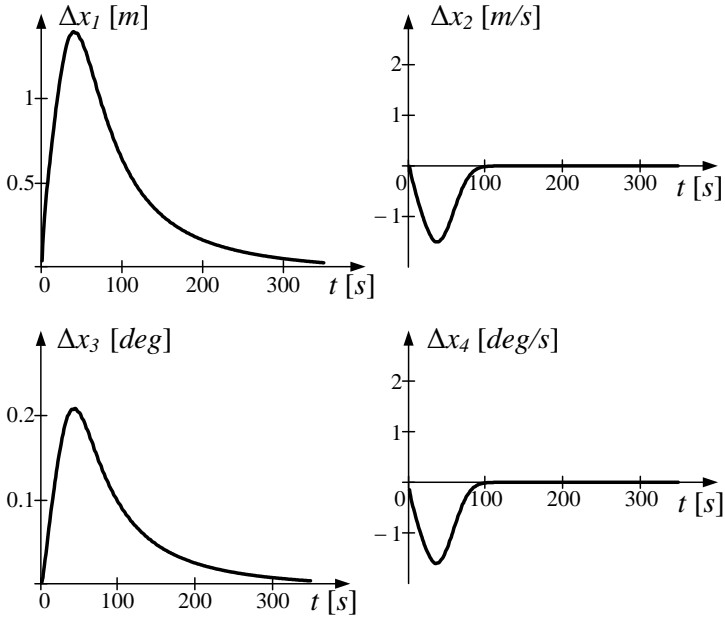

**Figure 19.** The graphs of the dynamics error.

When changing the external conditions for a large-sized object moving, it is necessary to re-start the identification process. The start clause of the identification phase is

$$|x_o(t) - x_m(t)| > \varepsilon. \tag{29}$$

When the outer conditions of bulky object motion change, it is needed to re-start the identification process. The condition for the activation phase identification

$$|y_o(t) - y_m(t)| > \varepsilon. \tag{30}$$

The study of the effectiveness of the procedure for identifying the parameters of a ship moving model on a slip by the recurrent method of stochastic approximation confirmed the adjustment problems of the algorithm associated with the choice of initial design values and weights that affect both its convergence and error and also revealed that the allowable time allotted on the evaluation of model parameters. When identifying model parameters for adaptive real-time control of the process of moving a large object, it is advisable to use the least squares method [17].

## 3. Methods

Computerized control systems are often introduced to manage complex technical complexes. A characteristic feature of such systems is the presence of several levels of control. To control the slip-raising complex, it is advisable to use a two-level control system. The first (lower) level is used for direct control of individual slip actuators, while the second, the upper level of the system, serves to implement a specific strategy for launching or lifting a vessel [18].

Before the start of the salvaging works, an obligatory check of all units and mechanisms of the slip is performed. At the next stage, the operator sets the slip operation mode (descent/ascent) through the appropriate SCADA window and selects the type of vessel to be prepared for installation on the slip carts (if the required type of vessel is not in the database, the operator manually enters the information).

After entering the data, the system processes them using the available models and displays the corresponding recommendations on the display. After installation of the vessel on carts and the beginning of their movement along the rail tracks, the control system continuously monitors the main parameters of the operating ship-lifting complex using the monitoring subsystem. The obtained data is used by local control systems. Operating parameters of the local control unit (LCU) are sent to decision support system, which serves the operator's workstation. The central control unit (CCU) determines the presence of deviations that are corrected by the LCU under the visual control of the operator. The launching or lifting of the vessel can be stopped at any time in the case of an emergency or as a result of a successful completion of work.

The algorithm of functioning of the computerized process control system for launching/lifting the vessel is shown in Figure 20.

The lower control level consists of the number of modules, the exact number of which depends on the number of slip carts. The slip trolley is driven by its electric drive, as a result, for the synthesis of control in the local system of the lower level, it is necessary to use information about the load (cable tension) and the operating parameters of the electric motor. Each of the lower-level control modules is a microprocessor system; however, due to the use of different electric motors on the slipways, they may have a different device.

The upper level of the control system is implemented using a controller, a set of appropriate input/output modules that form the central control unit (CCU), whose task is to coordinate the work of local lower level systems in order to stabilize the rotational movements of the vessel on the slip and maintain it uniform translational motion.

Effective operation of the central control unit is possible subject to the availability of an appropriate algorithm, a special database adequate to the mathematical model of the process of moving the vessel on the slip and timely information from the monitoring subsystem about the position of the trucks, their speed, the tension of the cable of each truck, the value of current in the electric motor circuits skewing of the vessel and from the control modules of the lower level.

To form the structure of information and measuring system, it is advisable to divide the factors that have a different disturbing effect on the work of a slipway, into four groups according to a functional feature, as shown in Figure 21.

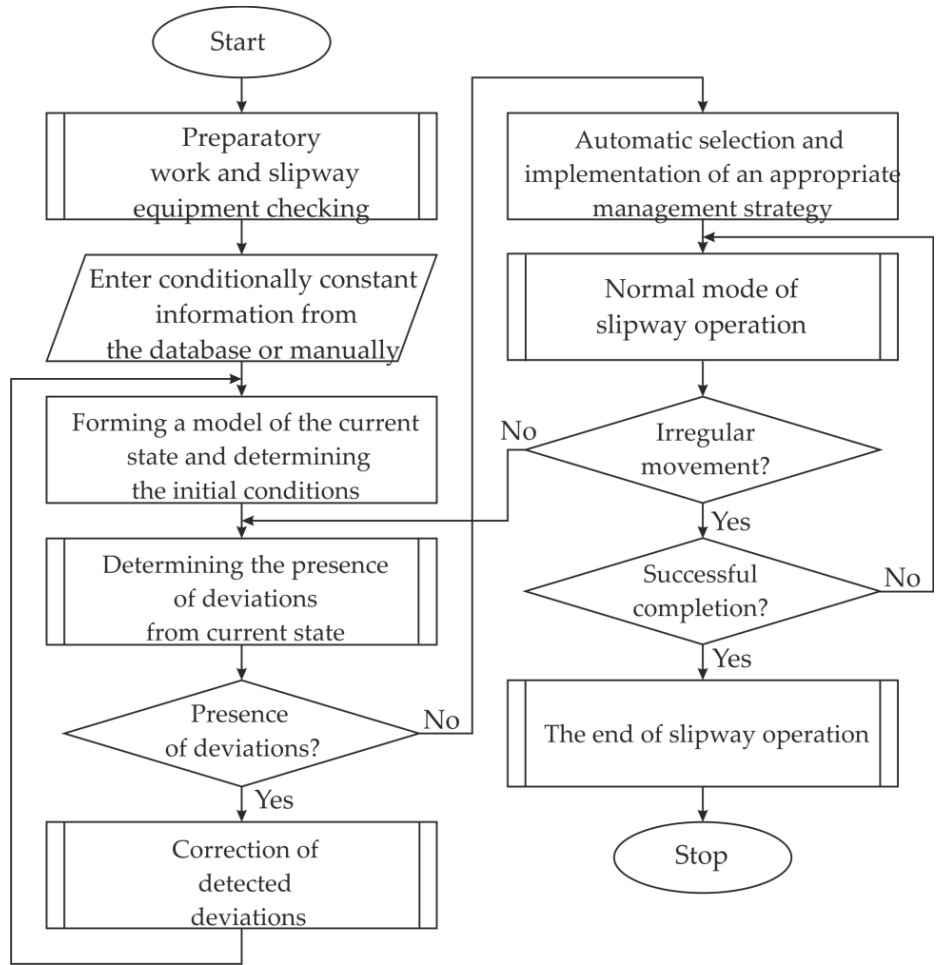

**Figure 20.** The algorithm of the control system work with the slip.

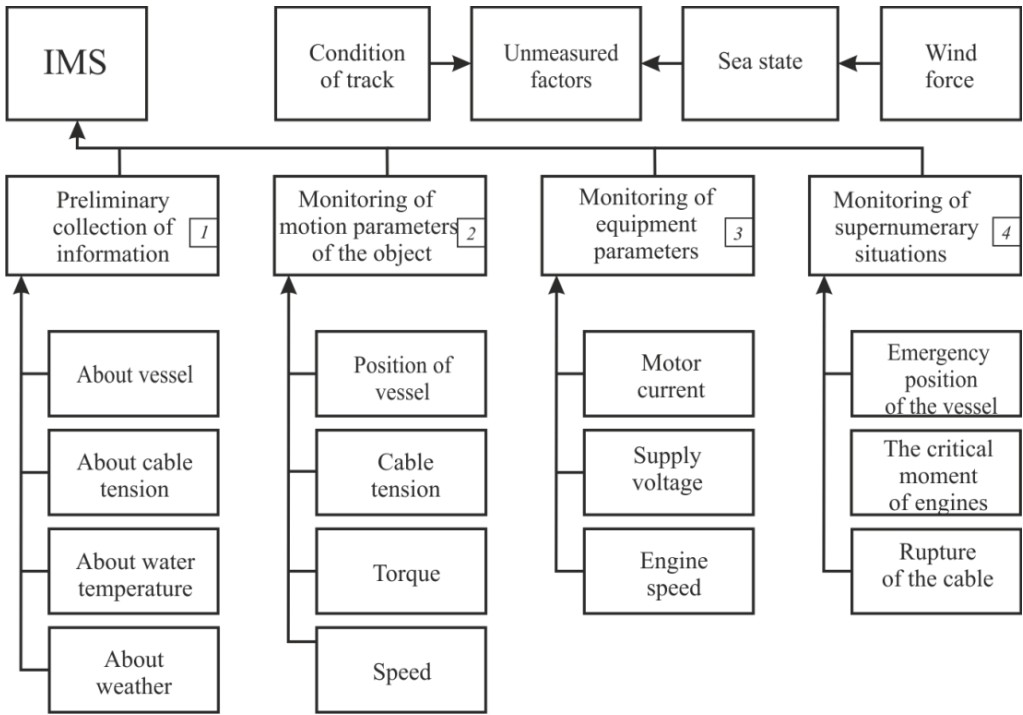

**Figure 21.** Analysis of factors affecting the work of slipway.

Before starting the procedure of launching the vessel, it is required to collect preliminary information about the object and the external conditions.

Information on the type of vessel, its weight and size characteristics and the distribution of weight loads should be set into the control system. The position of the carts should also be checked. Console ship-lifting complex should be provided with hardware and software automated workplace (AW). Continuous monitoring and control of the process parameters by the operator allows us to implement the SCADA-system. The developed models, theoretical and methodological approaches and recommendations can be used for the decision support system development as a part of the control system.

Decision Support Systems (DSS) includes the following components: a database (DB), a modelling subsystem, a problem solving subsystem, an interactive subsystem, a data transfer subsystem and an information processing subsystem.

Data on the type of vessel, dock drawing of the vessel, that can be developed by the technical department of the enterprise. The database should store information on the largest possible number of vessels that can be serviced by the shipyard, since the expected distribution of load between carts plays an important role and the coordinates of the centre of gravity of vessels, due to their construction, are different. The availability and development of such a database allows us not only to automate the slipway but also to reduce the time of preparation for ship-lifting or launch operations.

Information on weather conditions and their impact on the work of the slip should also be stored in the database. The ship should be lowered under favourable weather conditions. However, due to the limitations imposed by the time of construction/repair and delivery of the vessel to the customer, as well as due to the reduction of equipment downtime, it can be problematic to delay ship-lifting work; therefore it is necessary to control the weather factors in order to be able to launch the vessel under satisfactory weather conditions.

The modelling subsystem should contain the developed models of the main units of the slip and model the operation of the ship-lifting complex based on the use of data entered by the operator or obtained from the database, helping the operator to evaluate the progress of work and decide on the management of the ship-lifting complex using the task-solving subsystem. The data transfer subsystem connects the DSS with the information measuring subsystem and the lower level LCU. The interactive subsystem at the present stage is usually implemented by means of SCADA-system.

The structure of the DSS is shown in Figure 22.

The structural scheme of the computerized control system is presented in Figure 23. A set of sensors is installed on each electric drive of the slip cart: torque, cable tension and current and for extreme drives, in addition, range-finder sensors. The information collected by the sensors enters the pre-processing information units (IPPU). The sets of sensors and pre-processing blocks of the information form the monitoring subsystem. From each IPPU, information is transmitted to the appropriate local control system, as well as to the central control unit. A separate LCU consists of an information processing unit (IPU), which serves to receive and process data from the monitoring subsystem, an engine control unit (ECU) used to control the electric motor and a control module (CM) which controls the operation of the LCS, as well as the CCU, receiving commands from it and transmitting information about the functioning of the LCS. Local control systems form the lower level of control. The upper control level is represented by the central control unit and the automated workplace of the decision maker. The CCU is developed on the basis of one of the modern controllers that have proven themselves to be reliable and have qualified technical support, for example, on the basis of the Siemens SIMATIC S7-300 universal modular controller.

The controller of the CCU, on the basis of the collected information from all IPPU and all LCS, performs basic calculations in the system and then adjusts the work of the control modules. Using a SCADA system, the results of the CCU operation are visualized on the display screen of the operator. As a SCADA system for an automated workplace, it is possible, respectively, to use a multifunctional an d universal SIMATIC WinCC system.

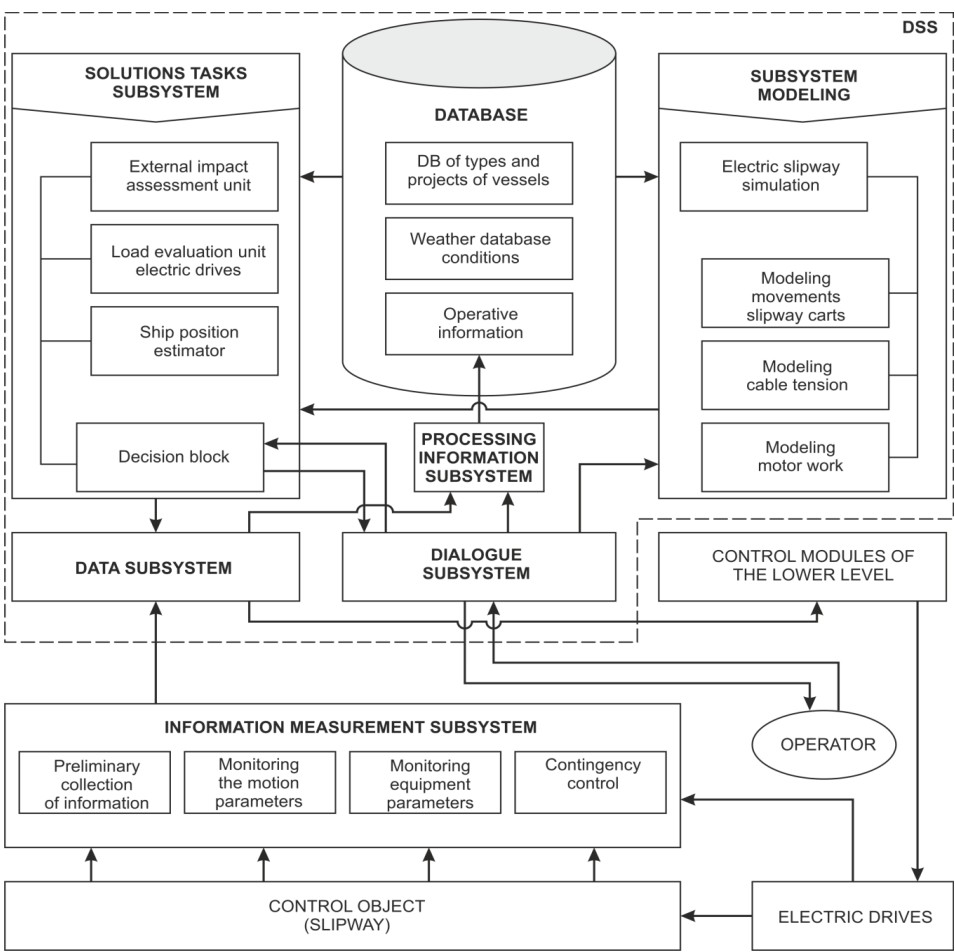

**Figure 22.** The structure of the DSS for the slip control system.

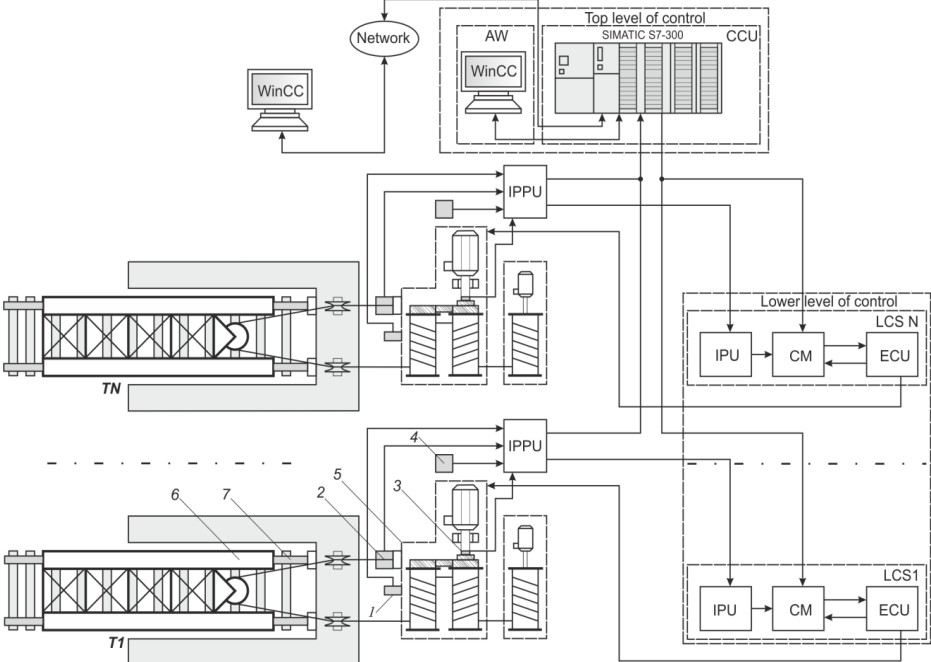

**Figure 23.** Block diagram of the computerized control system of the slip: 1—measuring the distance to the vessel, 2—sensors for measuring the tension force of the cable, 3—measuring the moment on the shaft, 4—sensors for measuring current, 5—electric drives, 6—ship-carrying carts, 7—rail tracks.

The considered computerized system will allow us to increase the reliability of the slip operation and to ensure increased safety during the execution of lifting operations under uncertainty about external and internal factors of a random nature.

## 4. Conclusions

Analysis of the process of moving the ship on the slipway revealed the following problems: the need for coordinated management of a group of electric drives, the frequent occurrence of emergency situations, the impact of external and internal factors of random nature. Ensuring safety and improving the reliability of the operation of the ship-lifting complex is possible through the introduction of an information management system and the use of adaptive management methods for the process of moving a ship on a slipway.

To analyse the operation of individual electric drives throughout the process of moving the vessel, a dynamic model of motor load based on the analysis of the forces acting on the ship's trolley was proposed. It allows estimating the cable tension at any point along the way depending on the design parameters of the system and external factors. To obtain an analytical description of the spatially changing system parameters, the apparatus of the theory of fuzzy sets was used.

A linearized model of the dynamics of the complex movement of the vessel in the process of moving on an inclined surface, taking into account the joint translational and rotational movements, as well as the structure of the multi-drive system, has been developed. The model, obtained takes into account the influence of external factors and changes in the parameters of the ship-lifting complex, allows us to ensure operational control and timely identifying the occurrence of critical situations during the operation of the ship-lifting complex.

The procedures for identifying the parameters of a ship moving model on a slipway by recurrent methods of stochastic approximation and least squares for use in an adaptive control system, that preserve the structure of the matrixes of the linearized model of the process of moving the ship on the slipway, are investigated.

The issues related to the implementation of the proposed models and methods for the operational control of the ship-lifting complex based on the use of modern microcontroller controls are considered.

The ways of further improvement of the decision support system in the automated system of operational management of the situation assessment module have been identified to ensure the coordinated operation of the multi-drive system, which allows us for the maximum automation of the operational management process of the ship-lifting slipway-type complex.

**Author Contributions:** H.R. and O.P. designed the model and the computational framework and analysed the data and carried out the implementation. A.O. performed the approbations. O.P. wrote the manuscript with materials from all authors. A.O. contributed to the final version of the manuscript. H.R. supervised of project.

**Conflicts of Interest:** The authors declare no conflict of interest.

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
