# Peer review of "Using Recurrent Procedures in Adaptive Control System for Identify the Model Parameters of the Moving Vessel on the Cross Slipway"

_data, 2018_

Round 1

Reviewer 1 Report

Major:

1.

Sections of this paper are not clear. Section 1 is "Summary", Section 2 is "Data" but content is different. "Data" section is related to "Model".

Analysis of other works is reduced to few sentences only in "Summary".

2.

This paper is related to the "Automation and Control" area directly and is not in the scope of this journal (https://www.mdpi.com/journal/data/about  - Aims section)

Minors:
Keywords - without "the" and "a"
l.284 Where <- where
l.289 ais <- a is  (space missing between alpha and 'is')
l.297, l.298 - 0,5 <- 0.5  use dot not a coma (English numbers !)
Eq.25 and others - use dot not a coma again
Fig.17,18  t,s  <-  t [s]   - it is more often used

Author Response

Our team of authors thanks you for your attention to our work and for constructive criticism.

Small remarks were fixed by us.

I would like to comment Your significant remarks as follows.

Our work is really closely related to the area of "Automation and Control". However, the methods of recurrent analysis formed the basis of our work. Our article presents the applied problem of our research (automation of the ship launching process) and some technical implementation in the form of a control system project. The article has been expanded to improve the perception of our research, however, it caused some inconsistency of article sections.

Reviewer 2 Report

The paper is related to adaptive control for identify the model parameters of the moving ship on the cross slipway. The topic is interesting to the academic community. My main concern about the paper is whether it actually makes a sufficient new contribution to the literature. Thus, the recommendation for publication hinges crucial on whether or not the authors can make a convincing case that this new paper makes a sufficiently distinct new contribution to the literature. At the moment, this is completely missing. In order to enhance the article quality, I suggest the following remarks be taken into account:

·       Line 27: ‘Introduction’ instead of ‘Summary’.

·       Lines 39-42: Please delete the numbers.

·       Figure 2: Please provide the meaning of notations in figure.

·       Line 76: This paper does a very poor job of literature review. The list of references is unacceptably incomplete. In order to be compared with other possible approaches, the authors should enrich their text with relevant references. For example, as:

-   „Adaptive system for steering a ship along the desired route” Mathematics vol. 6, no. 10(196), 2018 (1-11)

·       Line 79: In the last paragraph of the Introduction, material should be added regarding existing work carried out by other researchers and the way the authors is coming to improve or to excel these methods.

·       Line 251: ‘Figure 15’ instead of ‘Figure 3’.

·       Figures 16-17: The authors should add units.

Author Response

Our team of authors thanks you for your attention to our work and for constructive criticism.

Small remarks were fixed by us.

I would like to comment the remark about the significance of the work for literature as follows.

Our article is devoted to solving a specific applied problem (automating the process of launching a ship). We investigated in detail the small technological area (controlled launche of the vessel using a cross slipway). After collecting the experimental data, we used the already existing methods of recurrent analysis and the principles of constructing control systems. Thus the novelty of our work may be the following:

The dynamic model of load of the electric drive of the slipway cart is obtained. The model allows to analyze the influence of external factors and random influences during the whole process of launching / lifting of the vessel;

The mathematical model of the dynamics of a vessel movement in the process of launching / lifting in the state space is developed. The model allows to identify the mode of multi-drive system taking into account its structure.

The methods for assessing the parameters of the vessel during launch to water, which provide the opportunity to identify the mode of operation of the multi-drive system, based on the analysis of information from the subsystem of monitoring, were developed.

The subsystem of monitoring has been improved due to the use of new methods and tools for measuring the parameters of the slipway.

Round 2

Reviewer 1 Report

OK, but this paper is not related to "Data" journal in my opinion.

Reviewer 2 Report

The authors addressed my previous comments. In order to enhance the article quality, I suggest the remark be taken into account:

·       „The dynamic model of load of the electric drive of the slipway cart is … The subsystem of monitoring has been improved due to the use of new methods and tools for measuring the parameters of the slipway.” - The authors should add arguments to manuscript that this paper makes a sufficiently distinct new contribution to the literaturę.